# Imbalance between Employees and the Organisational Context: A Catalyst for Workplace Bullying Behaviours in Both Targets and Perpetrators

**DOI:** 10.3390/bs14090751

**Published:** 2024-08-27

**Authors:** Gülüm Özer, Jordi Escartín

**Affiliations:** 1Institute of Psychiatry, Psychology & Neuroscience, King’s College London, London SE5 8AF, UK; jorge.escartin@kcl.ac.uk; 2Department of Social Psychology and Quantitative Psychology, Universidad de Barcelona, 08035 Barcelona, Spain

**Keywords:** workplace bullying perpetrator, workplace bullying target, Job Demands–Resources model, quantitative and qualitative analysis, three-way model, correspondence analysis

## Abstract

Research on workplace bullying from the perpetrators’ perspective is limited, leading to few interventions tailored to them. This gap stems from insufficient understanding of how organisational conditions and individual dispositions trigger or amplify perpetrator behaviour. To develop effective interventions in preventing bullying, perceived organisational factors were examined. This study, guided by the Job Demands–Resources (JD–R) model and the three-way model of workplace bullying (WB), aims to explore the effects of perceived (im)balances in the task or employee focus, social atmosphere (positive or negative), and hierarchical structure (high or low) in organisations on the incidence of WB, both in terms of perpetrator and target experiences. The research involved 1044 employees from multiple Spanish organisations and sectors: 51.7% in services, 24.6% in education, 14.5% in wholesale and retail trade, and 9.2% in manufacturing. The average age of participants was 35.43 years (SD = 10.9), with a female majority of 61.3%. Using a cross-sectional study design, the experiences of being a target and perpetrator were quantitatively assessed through validated WB target and perpetrator scales. Additionally, the study qualitatively examined employees’ perceptions of their organisational context through free descriptions, using adjectives to depict their views. Correspondence analysis was employed to test the hypotheses, and the results affirmed them. The study found that perceived imbalances in organisational focus, social atmosphere, and hierarchy correlate with higher WB target and perpetrator reports, whereas balanced atmospheres correlate with no WB. This research underscores the significance of balanced organisational contexts in mitigating WB. It highlights a gap in the current literature regarding the broader organisational factors that influence bullying and advocates for a more in-depth understanding of these dynamics. Furthermore, it contributes to the existing body of knowledge by offering a comprehensive and harmonised approach to preventing WB, transcending isolated interventions.

## 1. Introduction

Historically, scholars have developed substantial knowledge about individual and personal reasons behind behaviour in general and organisational behaviour in particular. However, this behaviour is not only a result of persona but also stems from—and according to some psychologists, even to a bigger extent—context [1]. This latter idea has been widely applied in the field of workplace bullying (WB), in which bullying has predominantly been attributed to the organisational context [2,3]. WB is defined as negative behaviours targeted at an employee and their work systematically over a long time. Due to their nature, recurrence, systematisation, and damaging effects on the organisation and its employees, these bullying behaviours must be avoided or inhibited for good organisational functioning [4]. 

Scholars stressed the need for more theoretically driven studies to gain more insight into the antecedents of bullying [5,6,7]. In response, studies have applied the Job Demands–Resources model as a theoretical framework (JD–R model) [8]. Their results confirm the JD–R model’s assumption that combining high job demands and low job resources may trigger WB [9]. As such, these studies indicate that role stressors trigger stress and create the conditions for WB to emerge, where reports of WB target and perpetration increase [10]. 

Although these studies have shown promising results through quantitative data, we identify two gaps. First, the demands and resources investigated are predominantly located at the level of the employee’s task. Previous research to date has developed limited insight into antecedents on the level of the broader organisation regarding WB perpetration. Studies on the organisational environment revealed that absent managers and poorly designed organisational systems that neglect employee well-being foster conflict-prone settings. These environments lead to stress, exhaustion, and frustration, which can trigger bullying behaviour. When such behaviour is ignored or not condemned, it becomes normalised, even legitimised, leading to a cycle of increasing targets and perpetrators. However, some individuals, even without perceiving the environment as stressful, still engage in bullying, which requires further research to understand why [11]. This observation aligns with scholars [12], who called for more qualitative studies that contribute to knowledge about organisation-specific demands and resources that have been overlooked. Second, stress research has typically described stress and its reactions as a process characterised by a misbalance in what employees are to perform and the resources they have at their disposal to do so. Whereas imbalance is detrimental regarding stress reactions, balance protects employees from strain and even motivates them to reach their full potential [13]. Despite this well-known conceptualisation, studies have not explored this imbalance–balance idea concerning WB. 

The current study wants to address these voids by (a) specifically tapping into organisational antecedents of bullying, (b) drawing on a model that could help us define these antecedents while taking into account the perspective of imbalance versus balance, and (c) applying a more in-depth research methodology to unravel predictors of WB target and perpetration reports. 

### 1.1. Theoretical Approaches—The Three-Way Model

Our aims will be addressed by focusing on the propositions elaborated in the three-way model of WB [14]. The model has been built based on in-depth analyses by key informants—e.g., union representatives, managers, and persons with a confidence role within the organisation—of more than 80 bullying incidents. The model integrates the multiple causes—one being the organisation—that explain WB from the perspective of perpetrators and targets. The causes or antecedents can be linked to being a perpetrator or target of bullying through three processes (i.e., in ‘three ways’). The three-way model suggests that WB is caused by (1) inefficient coping with daily stress, (2) escalating conflicts, and (3) organisational culture, allowing such behaviours to exist.

First, antecedents could trigger bullying through the process of stress and frustration. Active-inefficient coping with stressors and frustration—i.e., by converting frustrations into negative acts towards a co-worker (Revised Frustration Aggression Theory; [15])—may trigger the enactment of WB as a perpetrator. Passive-inefficient coping—i.e., by withdrawing from the source of the strain and violating expectations at work—may cause others to adopt a negative attitude towards the strained employee, justifying negative behaviour such as bullying as a means to restore order (Social Interactionism; [16]). As a consequence, the strained employee may become a target of bullying. 

Second, the antecedents could trigger bullying through personal conflicts in which conflicts escalate into bullying when using escalating conflict management styles. In this case, the powerful employee in the conflict may become the perpetrator; the powerless employee may become the target. 

Third, job, team, and organisational antecedents could trigger bullying by allowing or encouraging bullying through social norms and climate, reflected in artefacts such as policies, reward systems, or managerial decisions.

### 1.2. Theoretical Approaches—Job Demands and Resources Model

According to the JD–R model [8], any work environment can be classified into two general categories: job demands and resources. Job resources are those physical, social, or organisational aspects of the job that are “functional in achieving work-related goals, reduce job demands and the associated physiological and psychological processes, and stimulate personal growth, learning, and development” [17]. For job resources to impact employee attitudes and behaviours, there is a need for challenging demands [18]. Typical examples are autonomy, social support, or performance feedback. Job demands are defined as those physical, social, or organisational aspects of the job that require sustained physical and psychological costs. 

According to the health impairment process, job demands may exhaust employees’ resources and lead to energy depletion and health problems [19]. A point of departure of leading stress models such as the JD–R model is that job strain results from a disturbance of the equilibrium between the demands employees are exposed to and the resources they have at their disposal. High demands and low levels of resources result in the highest levels of strain and the lowest level of health and well-being [20]. When the organisational environment does not offer the necessary resources and challenges, basic human needs cannot be fulfilled [21], resulting in cynical and potentially negative attitudes towards work (such as apathy and low morale) [19], and thus the disturbance of the equilibrium mentioned above. 

Whereas the JD–R model stresses the importance of demanding but challenging jobs, the opposite is also true, and resources are not only necessary to deal with job demands but also important in their own right [22]. High job resources can lead to boredom, low arousal, and negative emotions associated with insufficient stimulation, which has been related to counterproductive work behaviours [23]. 

Organisational contexts where the emphasis is on maintaining relationships but not necessarily getting things done are prone to foster a cosy environment, resulting in a disturbance of equilibrium. However, when job demands and resources are high, employees are expected to develop average strain yet high motivation, leading them to become very productive [24]. 

Scholars [22] showed that employees had positive work attitudes when job demands and resources were both high. This equilibrium or balance represents a profit for organisations and employees. It is also in line with the effort–recovery model [25], which stresses that employees are willing to dedicate their efforts and abilities to the work task when the organisational environment offers resources to them. 

As these theoretical predictions have rarely been the specific focus of research, scholars [12] have claimed the need for future studies that not only focus on a disturbance of the equilibrium between demands and resources but also analyse the consequences of equilibrium between them, especially when having both demands and resources on an elevated level. Additionally, previous research has shown that if employees had task autonomy under high job demands, they still reported WB perpetration [26], highlighting the need for further studies on balance and imbalance situations. 

### 1.3. Theoretical Approaches Combined—The Dimensions of Imbalances Created by Organisations Triggering WB

The three-way model [14] was developed by collecting data from 19 Belgian organisations across different sectors and interviewing 126 key informants, including union representatives, HR managers, and employees with roles related to WB. These informants provided detailed descriptions of 87 WB incidents. The study’s qualitative approach allowed for a comprehensive understanding of the dynamics behind WB, revealing the potential role of three organisational factors as antecedents (i.e., organisational focus, social atmosphere, and hierarchy). It suggested three dimensions that may cluster organisational antecedents of WB, both from the perpetrator’s and target’s perspective. The two poles that reflect an opposite condition are key in these dimensions. Based on the in-depth case analyses [14], the model elaborates that an imbalance—or focus on one of the poles of the three dimensions—will promote WB. An imbalance in these dimensions would encourage bullying as they contribute to at least one of the three explanatory processes (see Figure 1): (i) they trigger frustration, (ii) lead to conflicts, or (iii) allow negative acts and bullying when the end justifies the means.

Such imbalance is comparable to the disturbance of the equilibrium stressed within the JD–R model [22] and the equilibrium at low levels between demands and resources within the JD–R model [23]. A balance—a focus in the middle between these poles—will discourage WB; such balance is comparable to the equilibrium between high levels of both demands and resources within the JD–R model [22]. 

In contemporary organisational settings, WB is persistently and adversely affecting employee well-being, productivity, and overall organisational health. Human resource practitioners are in need of new approaches to intervene in situations triggering WB, and especially need to focus on WB perpetrators and what triggers them. Understanding the nuances of how organisational focus impacts employee interactions and workplace culture is critical. This exploration can provide valuable insights for managers and HR professionals aiming to design interventions that promote a positive organisational climate and reduce the prevalence of WB.

The model in this research describes the processes leading to WB perpetration and target based on a perceived imbalance in organisational (1) focus, (2) social atmosphere, and (3) power and hierarchy, replicating the model used by previous researchers [27]. Therefore, the present study examines the influence of the perceived balanced or unbalanced organisational context (i.e., the organisation’s focus, social atmosphere, and hierarchy) on employees’ reports as a WB perpetrator and target). These relationships will be approached qualitatively based on the three-way model (see Figure 1).

#### 1.3.1. Organisational Focus

The first dimension in the three-way model refers to the organisation’s focus, with one pole being task-focused and the other being employee-focused. Examples of antecedents belonging to this dimension are a highly goal-oriented organisational culture with a task-oriented management style and task fragmentation towards piecework. Based on the case analyses by key informants, the authors conclude that goals that merely focus on organisational needs may encourage bullying from the perspective of the perpetrator and the target [14]. This is reaching the organisation’s goals without taking care of any other needs that may be important, such as career perspectives for the employees to develop (i.e., having low resources and high demands as per the JD–R model, H1a). Alternatively, organisational circumstances that overfocus on employee needs without balancing these with organisational goals may also trigger WB to emerge (i.e., having low demands and high resources as per the JD–R model, H1b). Organisational settings that discourage bullying would typically balance between focus on the tasks and employees. These settings will not focus on one pole yet position themselves in the middle of the dimension, balancing the two poles and cultivating a supportive environment (i.e., having high resources and high demands as per the JD–R model, H1c). 

#### 1.3.2. Organisational Atmosphere

The second dimension refers to the quality of the organisation’s general social atmosphere; many personal conflicts may emerge when there is a hostile atmosphere or a lack of clear boundaries regarding work and private lives. As analysed by the key informants in the study [14], social climates characterised by very little personal contact between the employees or that are very hostile may encourage incidents of WB (i.e., having low resources and high demands as per the JD–R model, H2a). These organisational atmospheres would be restricted mostly to interactions necessary to serve individual goals instead of pleasant human contact. Alternatively, social climates in which workers draw a very thin line between work and private matters may encourage being a perpetrator or being a target of bullying too (i.e., having low resources and low demands as per the JD–R model, H2b). Again, a balanced social atmosphere that respects human relations and does not fade the line between work and personal matters would decrease WB as it would be a positive, supportive environment, reducing conflicts (i.e., having high resources and high demands as per the JD–R model, H2c). 

#### 1.3.3. Organisational Hierarchy

The third and final dimension refers to power relations and hierarchy within the organisation, being too much versus too little. Antecedents belonging to this cluster could be a power vacuum within the organisation, laissez-faire leadership, or bureaucracy. Specifically, too much power distance or hierarchy may create an environment in which problems are prone to persist, which in the end may lead to incidents of WB (i.e., having low resources and high demands as per the JD–R model, H3a). Too little power distance or too little hierarchy within the organisation creates a context in which the tools to efficiently respond to problems are lacking, ultimately encouraging being a perpetrator or a target of WB (i.e., having low resources and low demands as per the JD–R model, H3b). Again, a balanced hierarchy within the organisation—not too much or too little—is suggested to discourage bullying (i.e., having high resources and high demands as per the JD–R model, H3c).

#### 1.3.4. Research Hypotheses

Previous researchers [28], used Social Exchange Theory to explore under-theorised aspects of WB, focusing on organisational justice, psychological contract breaches, and perceived organisational support. They emphasised the subjective nature of bullying, where perceptions of injustice significantly impact both targets and bystanders. Therefore, organisational contexts that reflect an injustice and unbalance among the three dimensions (i.e., organisational focus, social atmosphere, and hierarchy) are expected to foster negative employees’ behaviours (see Figure 1). 

According to the health impairment process from the JD–R model [18], these organisational characteristics could provide a necessary or appropriate context for bullying behaviours to emerge in an organisation’s workforce. As suggested by previous researchers [19,21,23], such imbalance could drive different levels of strain and motivation (i.e., causing high demands—low resources, low demands—low resources, and high resources—low demands), leading to negative work attitudes.

On the contrary, organisational contexts that reflect a balance among the three are expected to minimise negative employee behaviours, leading to average strain yet high motivation and positive work attitudes [21,22,24]. Therefore, we hypothesised the following:
**Hypothesis** **1.***Perceived task or employee-oriented focus within the organisation will be associated with a high degree of WB target and perpetration reports (H1a, b), whereas a balance-oriented focus will be associated with a low degree of WB target and perpetration reports (H1c)*.
**Hypothesis** **2.***A perceived hostile or too informal atmosphere within the organisation will be associated with a high degree of WB target and perpetration reports (H2a, b), whereas a perceived balanced or positive atmosphere will be associated with a low degree of WB target and perpetration reports (H2c)*.
**Hypothesis** **3.***A perceived high or low hierarchy within the organisation will be associated with a high degree of WB target and perpetration reports (H3a, b), whereas a balanced hierarchy will be associated with a low degree of WB target and perpetration reports (H3c)*. 

## 2. Method

### 2.1. Participants

In the present study, 1044 employees from 54 organisations in Spain participated, where undergraduate Organisational Psychology students at the University of Barcelona approached them (Table 1). 

The collection of information about the sample took place through paper-and-pencil questionnaires and was carried out during working hours in the organisations’ facilities that provided access to the participants. The response rate was 79%. The average age of the sample was 35.43 (*SD* = 10.91). About 61.3% of workers were female, and 50.1% were married or living together, while 41.3.% were single, 7.7% were separated, and 1% were divorced. The participants with higher education were 47.7%, and 70.9% had permanent contracts. Supervisors comprised 16.5% of the sample, and 17.4% of the sample earned more than €30,000. 51.7% of the total participants worked in service settings, 24.6% were from education, 14.5% from trade, and 9.2% were from the industry setting. 

### 2.2. Measures

Data were collected using a questionnaire consisting of two sections: a qualitative and a quantitative, where all the questions were administered in Spanish. The first part included qualitative questions tapping into organisational factors based on a description of the organisation. The respondents were asked to describe their company with a maximum of six adjectives (i.e., positive adjectives such as enjoyable or friendly and negative adjectives such as misleading or passive) (see Table 2). This methodology has been used in several studies [27,29] and has shown to be helpful when explaining individuals’ perceptions and actions concerning what is central, enduring, and distinctive about an organisation’s identity [30,31]. Specifically, the free-response format allowed participants to use their vocabulary to describe how they view their organisational context, maximising the expression of individual variance. 

The quantitative part included questions regarding the sociodemographic of participants and WB target and perpetrator reports. Five-point Likert scales were employed to measure all variables. The scales ranged from (0) strongly disagree to (4) strongly agree.

WB perpetration was measured using the 7-item NAQ-Perpetrators Questionnaire [32]. A sample item is “I have ignored, excluded or physically isolated others”. For the full scale, the mean score was 0.22 (SD = 0.39), and the Cronbach’s alpha was 0.83. All measures were administered in Spanish and respondents were assured of confidentiality and informed that they could withdraw from the study anytime.

WB target experience was measured using the behavioural experience method through the 12-items scale Escala de Abuso Psicológico Aplicado en el Lugar de Trabajo [33]. A sample item is “I have been excluded from celebrations and social activities organised by my co-workers”. For the full scale, the mean score was 0.24 (*SD* = 0.41), and the Cronbach’s alpha was 0.90.

### 2.3. Procedure

The participation of all employees was completely voluntary, and their anonymity was ensured. The qualitative descriptions of the organisations offered by participants were categorised by two academic coders, following three criteria, which are in line with the research objectives: (1) task-oriented, task/employee balanced-oriented, or employee-oriented focus; (2) negative, balanced/positive, or highly informal atmosphere; and (3) too little hierarchy, balanced, or too much hierarchy. For each of these categories, a “1”, “2”, and “3” were assigned, according to the description of the organisation (e.g., negative atmosphere = 1; balanced/positive atmosphere = 2; and highly informal atmosphere = 3). 

Two researchers independently classified the adjectives into four groups based on the three dimensions (i.e., focus, atmosphere, hierarchy levels), suggesting “balanced” and “unbalanced” environments. To ensure the reliability of the coding process, Cohen’s Kappa (k) was used to calculate the level of agreement between the two coders, which is a measure beyond that agreement expected to occur by chance for each of the three dimensions. The measure of agreement for organisational focus was 0.86 (*p* < 0.001), for organisational atmosphere was 0.87 (*p* < 0.001), and for organisational hierarchy was 0.85 (*p* < 0.001), suggesting high agreement [34]. Disagreements between coders were settled to confirm the reliability further. WB perpetration and target scores were categorised into low-, medium-, and high-intensity experiences, according to the frequency of negative behaviour experiences [35].
-If participants reported one behaviour once a week or almost daily, they were labelled as having high perpetration or target experiences (=3).-If participants reported one behaviour once a month, they were labelled as having medium perpetration or target experiences (=2).-If participants reported one behaviour now and then, they were labelled as having low perpetration or target experiences (=1).-Participants were labelled uninvolved if they did not report any behaviour (=4).

### 2.4. Data Analysis

We conducted a correspondence analysis (CA) to analyse the association between the organisational dimensions (focus, atmosphere, and hierarchy) categorised into three levels and the different degrees of WB target and perpetration reports categorised into four levels. CA is a statistical technique to explore and visualise relationships between categorical variables in a contingency table. The technique reduces the dimensions of the data. CA is typically applied to contingency tables, where the tables show how categories of one variable are distributed across categories of another variable. The primary output of CA is a graphical representation as a biplot that visually depicts the associations between the categories of the variables. This visualisation helps to identify similarities and differences between categories. In the resulting plot, categories close to each other are similar, indicating a relationship. Categories that are far apart are dissimilar [36]. We used CA because it has shown to be the most appropriate tool for capturing the covariance relationships between different variables, regardless of their typology and sample size. Thus, the outputs performed by the CA are in the form of coordinate graphs; within these graphic representations, the physical closeness is interpreted as a covariance relationship. Data were analysed using SPSS 26 (IBM Corp, New York, NY, USA). 

## 3. Results

### 3.1. Descriptive Statistics

Table 3 shows the means, standard deviations, and inter-correlations between variables under the study. 

WB target reports decreased as age increased, while reports of WB perpetration were higher among males, and being a target of WB was highly related to being a perpetrator. It is also worth noting that most supervisors were males, but being a supervisor was not associated with getting involved in WB [37].

When WB experiences were categorised and mapped by intensity (Table 4), 295 participants (28%) were not involved in any negative experiences. While 20.2% of the participants reported varying degrees of WB target experiences without enacting WB, 14.4% reported varying degrees of WB perpetration without being a target of bullying. The majority of the participants, 37.2%, were both enacting and being a target of WB. 

When the targets were asked about who the bully was, 116 participants (11.1%) replied and reported that 62.9% of them were bulled by their supervisors, 27.6% by their colleagues, 6.9% by both their supervisors and colleagues, and 2.6% reported that they were bullied by their subordinates, suggesting upwards bullying, which was the least common within the participants. 

### 3.2. Correspondence Analysis on Hypothesis

Among 1040 participants, 750 (72.1%) provided 3381 adjectives in Spanish to describe their organisational environments. The correspondence analysis (CA) was conducted with WB perpetration and target intensities (i.e., low, medium, high, and no involvement in WB) and three dimensions of the organisational environment (i.e., focus, atmosphere, and hierarchy) with three distinct categories (too much, too little or balance), separately to test the hypotheses. When numerous combinations of factors are at play, a picture is easier to interpret rather than a number, and correspondence analysis provides graphical representations showing dissimilarities where the distance between the points is the key element of interpretation [38]. 

The first hypothesis was to analyse whether perceived organisational focus indicated an imbalance (i.e., too much focus on tasks or too much on employees) that would be associated with a high degree of both WB perpetration (H1a) and target (H1b). In contrast, a balanced focus will be associated with low WB target and perpetrator reports (H1c). The model summaries can be distinguished for perpetrators and targets. For perpetrators, the first axis (balance vs. imbalance) explained 92% and the second axis (intensity of WB) 8% of the variance. For targets, the first axis explained 98% and the second axis 2%. As can be seen in the CA coordinate map (Figure 2a), the vertical axis discriminated between the perceived balanced (right side of the graphic) versus imbalanced (left side of the graphic) organisational focus. 

We observed a clear association between a balanced focus and no reports of WB target experiences. Task focus was associated with high target reports, and employee focus was associated with low and medium target reports. Similarly, a balanced focus on employees and tasks was associated with no involvement in WB perpetration. In contrast, too much task or too much employee focus was related to varying degrees of WB perpetration (Figure 2b). Therefore, H1a and b were confirmed. However, as a balanced focus is associated with an absence of WB as opposed to a low report of WB experiences, H1c was rejected. 

The second hypothesis of this study was to analyse whether the organisational atmosphere, which was perceived as imbalanced (e.g., too hostile or too informal as to blur the divide between work and private lives), would be associated with high levels of perpetration (H2a) and target incidences (H2b). In contrast, a balanced atmosphere would be associated with lower WB perpetration and target experiences (H2c). The model summaries show that the first axis explained 98% of the variance, and the second axis explained 2% for target experiences; for perpetration experiences, the first axis explained 88%, and the second axis was 12%. As can be seen in the CA coordinate map (Figure 3a), the vertical axis discriminated between the perceived balanced (right side of the graphic) versus the imbalanced (left side of the graphic) organisational atmosphere. 

We observed a clear association between a balanced atmosphere and the absence of WB targets. A negative atmosphere was associated with medium and high target experiences, while overly informal settings were associated with low target experiences. Regarding reports of WB perpetration, a balanced atmosphere was associated with no involvement in WB perpetration. In contrast, a negative atmosphere was associated with medium and high reports of perpetration, while an overly informal climate was associated closely with low and medium levels of perpetration incidences (Figure 3b). Therefore, H2a and b were confirmed. However, as a balanced atmosphere is associated with no WB target and perpetrator reports as opposed to low, H2c was rejected.

The third and final hypothesis of this study was to analyse whether the characteristics of the organisation, when perceived by employees as much or little hierarchical (imbalanced), would be associated with a high degree of perpetration (H3a) and WB target experiences (H3b). Conversely, when employees perceive the organisational structure and power relations as balanced, it would be associated with a low degree of all study outcomes (H3c). According to the model summaries, the first axis explained 99.7% and the second axis 0.03% of the variance for target experiences; for perpetration experiences, the first axis explained 90% and the second axis 10%. As can be seen in the CA coordinate map (Figure 4a), the vertical axis discriminated between the perceived balanced (left side of the graphic) versus imbalanced (right side of the graphic) organisational hierarchy. 

We observed a clear association between a balanced “structure and power relationships” in the organisation with the absence of WB target experiences. However, a hierarchy that is too high was associated with a low level of being a target of bullying, and too little hierarchy was associated with medium and high WB target experiences. As for reports of WB perpetration, balanced power and hierarchy were associated with no involvement in WB perpetration. However, too little hierarchy was associated with a medium level of WB perpetration reports, while too high of a hierarchy was associated with low and high WB preparation (Figure 4b). Therefore, H3a and b were confirmed. However, as a balanced hierarchy was associated with no WB target and perpetrator reports as opposed to low, H3c was rejected.

### 3.3. Correspondence Analysis of WB Experiences

We categorised the participants into four categories (Table 5):-Target not a perpetrator—if they reported being subject to high, medium, or low bullying but did not report being involved in WB perpetration.-Perpetrator not a target—if they reported bullying others but were not bullied by others.-Target perpetrator—if they reported being subject to bullying and bullying others.-Uninvolved—if they reported that they were not involved in any WB experiences.

We observed that 14.4% of the participants were WB perpetrators but were not being bullied, whereas 20.2% were the target of bullying but were not bullying others. 

We used CA to understand if specific WB actors were clustered in specific organisational environments. 

When organisational focus was analysed (Table 6), we observed that the largest group of employees in the balanced focus environment (n = 352, 47%) were the employees who were not involved in WB (n = 125, 17%). Nevertheless, a balanced focus environment still harboured some targets, perpetrators, and target perpetrators. 

However, when the biplots were analysed (Figure 5), we noted that the first axis explained 98% and the second axis explained 2% of the variance. As can be seen in the CA coordinate map (Figure 5), the vertical axis discriminated between the perceived balanced (right side of the graphic) versus imbalanced (left side of the graphic) organisational focus. Therefore, a balanced organisational focus on tasks and employees was associated not only with “uninvolved” employees who did not report any WB experiences but also with the “perpetrator not target” group. Having a task focus was related to the “target not perpetrator” group, and employee focus seemed closely related to the “target perpetrator” group.

When the organisational atmosphere was analysed (Table 7), we observed that the largest group of employees in the balanced atmosphere environment (n = 428, 58%) were the employees who were not involved in WB (n = 153, 21%). Nevertheless, a balanced atmosphere still harboured some targets, perpetrators, and target perpetrators. 

When the biplots were analysed (Figure 6), we noted that the first axis explained 99% and the second axis explained 1% of the variance and the vertical axis discriminated between the perceived balanced (right side of the graphic) versus imbalanced (left side of the graphic) organisational focus. A balanced organisational atmosphere was associated with the “uninvolved” and “perpetrator not target” group. A negative atmosphere was closely related to the “target perpetrator” group, while having a too informal atmosphere was associated with the “target not perpetrator” group.

When organisational hierarchy was analysed (Table 8), we observed that the largest group of employees in the balanced hierarchical environment (n = 407, 54%) were the employees who were not involved in WB (n = 143, 19%). Nevertheless, even a balanced hierarchical environment harboured targets, perpetrators, and target perpetrators. 

When the biplots were analysed (Figure 7), we noted that the first axis explained 98% and the second axis explained 2% of the variance and the vertical axis discriminated between the perceived balanced (left side of the graphic) versus imbalanced (right side of the graphic) organisational hierarchy. Therefore, a balanced organisational hierarchy was closely related to the “uninvolved” group and the “perpetrator not target” group. Too little hierarchy was associated with the “target perpetrator” group, and too much hierarchy with “target not perpetrator” group.

In summary, the study found that WB target reports decreased with age and perpetration was higher among males, with a strong correlation between being a target and a perpetrator of WB. Among the participants, 28.3% were not involved in any negative experiences, while 20.2% reported being targets without enacting WB, and 14.4% reported perpetrating WB without being targets. The majority, 37.2%, were both targets and perpetrators of WB. When asked about the bully, 62.9% of the targets identified their supervisors, 27.6% their colleagues, 6.9% both supervisors and colleagues, and 2.6% their subordinates, indicating upwards bullying. A balanced focus on employees and tasks, as well as a balanced organisational atmosphere and power structure (hierarchy), was associated with the absence of WB experiences. Conversely, imbalances—whether in task or employee focus, atmosphere, or hierarchy—were linked to varying degrees of WB perpetration and target experiences.

The study also analysed which types of actors are clustered in different organisational environments and found that a balanced organisational focus, atmosphere, and hierarchy were associated with the highest proportion of employees not involved in WB. In a balanced focus environment, 47% of employees were uninvolved in WB, while balanced atmosphere and hierarchy environments had 58% and 54% uninvolved employees, respectively. Out of the 1044 employees, 293 worked in completely balanced environments in terms of focus, atmosphere and hierarchy simultaneously. However, still 48 (16%) of them were perpetrators not targets, and 186 (63%) were target perpetrators.

## 4. Discussion

The present study yields some innovations as the results further our understanding of WB—both from the perpetrators’ and the targets’ perspectives—and add to previous studies in the field [14]. Specifically, the current study not only contributes to existing knowledge by investigating organisational antecedents of bullying following recommendations made in the JD–R model’s literature [12] but also by focusing on perpetrators, an aspect that has long been a “black hole” in the WB literature [6,39]. 

This study aimed to provide empirical evidence to establish the importance of organisational characteristics, as perceived by employees, on WB. The results showed significant associations between a balanced organisational context and the absence of WB. The hypotheses that suggested higher reports of WB experiences (both as a target and a perpetrator) in an imbalanced organisational context were confirmed. However, the hypotheses that suggested a low intensity of WB would be related to a balanced organisational context were rejected as such environments were strongly associated with no WB involvement, as supported through the qualitative analyses performed by the correspondence analyses.

Too much task focus [27], too much hierarchy, and a lack of formal organisational policies that balance work and private life may create a mismatch of demands and resources [40], create stress, and lead to perceptions of being bullied. Therefore, organisations with a high hierarchy or too focused on tasks might need strategies to mitigate high reports of WB. 

On the other hand, too much employee focus and too little hierarchy may create a permeable boundary of work procedures, reducing trust in the organisation and leading to a negative atmosphere, laying the grounds for a stressful organisational environment. In such an environment, employees may perceive injustice, feel bullied [28], and act out to defend their positions [41]. Many previous studies showed that being bullied is associated with perpetration over cross-sectional and longitudinal studies [11]. Accused bullies also report distress and that they are being bullied [42]. Following the enactment of bullying, as conflicts may arise, bullies become more likely to become exposed to bullying themselves [43], which suggests a vicious cycle of perceptions of bullying and the enactment of bullying. 

These findings are consistent with social exchange theories, which stress that employees are more likely to elicit negative behaviours when their company does not support them, and they perceive that their organisation is not treating them well [19]. According to this approach, employees who perceive that the organisation they work for is not investing in their well-being are more likely to reciprocate through negative behaviour [19,32], showing implicit reciprocity norms that exist between employees and their employers [44]. Our results corroborate the existence of these implicit reciprocity norms and the power of specific organisational characteristics that increase negative organisational behaviour. 

One of the most fascinating insights of the present study is related to the proposition formulated by scholars [12], who claimed the need for a further understanding of the consequences of equilibrium, particularly because of having both demands and resources elevated. The present study has presented initial evidence showing that employees are expected to generate no negative behaviours when job demands and resources are high and balanced [24]. Therefore, this balance represents a competitive advantage for organisations and their employees, especially in light of flow theory, which posits that high levels of skills and demands lead to flow [45,46], an optimal experience that is considered a positive psychological construct that is important for organisations and beneficial for individuals regarding well-being. 

Although not hypothesised, further correspondence analysis on WB actors and organisational context revealed significant associations. Balanced environments in terms of focus, atmosphere, and hierarchy were all associated with the absence of WB, yet a small group of employees, classified as “perpetrators but not targets” and “target perpetrators”, survived in these environments. A previous longitudinal study also showed that perpetrators’ occupational status remains unchanged over time, where they do not face significant risks of being expelled or becoming unemployed [47]. 

### 4.1. Limitations and Future Research

The study’s design implies some constraints that may affect the interpretation of the results and possibilities for generalising our findings. Researchers should be wary of interpreting our results in a causal way owing to the cross-sectional design of our data. A related issue concerns social desirability, which may reduce the likelihood of obtaining accurate responses relating to bullying reports, particularly from the perspective of perpetrators. If social desirability is operating, this implies an underestimation of effects owing to a lack of variance. That is, relationships may become even stronger when accounting for social desirability. Nevertheless, several research focused on the perspective of the perpetrators has shown encouraging results regarding their use [41,48]. Future research should examine whether different forms and behaviour patterns characterise bullying within organisations according to the different organisational characteristics. 

### 4.2. Theoretical and Practical Implications

These findings support a conceptualisation of bullying antecedents that go beyond individual or interpersonal characteristics to one that includes the organisational context [49]. This study reinforces the theoretical premises of the three-way model [14], which argued that balanced workplace settings could avoid the reproduction of bullying behaviours. It aligns with the organisational environment hypothesis [35], which sees work conditions as a bullying trigger. Therefore, according to Salin’s bullying framework [50], workplaces can give rise to enabling, motivating, and precipitating processes for bullying. Organisations with a balanced/positive organisational context are unlikely to provide fertile soil for bullying because they would minimise and remove precipitating stressors. From an empirical perspective, the results are coherent with previous studies that found that such contexts can influence employees’ behaviours, health, and well-being. Previously, researchers [27] found that balanced organisational contexts fostered positive behaviour (i.e., altruism behaviour) and well-being (i.e., work engagement and job satisfaction). Therefore, it contributes to our understanding of several phenomena and significantly adds information to and builds on previous studies examining relational correlates of organisational context and employee behaviour and well-being. 

These findings have important implications for developing intervention strategies, as several behavioural and health and well-being outcomes are also raised from the organisational level. Managers should regularly assess the organisational context to identify potential issues early, such as assessing the perceived balance in organisational focus on tasks versus employees and the organisational hierarchy being a low versus high level and ensuring that neither side is extreme. Organisations could establish clear work procedures and ensure that managerial support is perceived as fair, consistent, and sensitive to stress prevention and promoting health and safety initiatives [51]. Additionally, organisations should ensure that the organisational atmosphere is positive without hostility and overly informal, breaching the boundaries of private lives. Management may establish policies that promote a supportive organisational environment with work–life balance. Since perpetrators often do not face significant consequences and may remain in their roles, organisations should address bullying from the perpetrator’s perspective as well. Leaders may include implementing interventions that focus on changing the behaviour of perpetrators, addressing the organisational factors that enable such behaviour and developing conflict resolution mechanisms that can address issues before they escalate into bullying. 

Finally, organisations can develop a flow state among employees by maintaining high job demands and resources, leading to higher productivity and well-being. This balance can serve as a competitive advantage, enhancing overall organisational performance.

## 5. Conclusions

The study advances the understanding of organisational environments and their role as enablers of WB experiences by examining them from both the perpetrators’ and targets’ perspectives, addressing a gap in the literature with an original methodological approach. 

An organisational setting that maintains a balanced focus on both its employees and tasks—where the organisation is well-organised, coherent, supportive, empathetic, and respectful of human relations—energises and motivates employees, leading to high performance even in the face of significant work demands. The research challenges the narrow focus on high demands causing strain, showing that high demands can be beneficial for employees when supported by adequate resources. Such balanced organisational contexts are associated with the absence of WB. Conversely, organisational settings that are disorganised, unmotivating, or controlling and exploitative of their employees miss the opportunity to unlock their employees’ potential. Imbalanced contexts lead to higher reports of WB, while balanced environments show no involvement in bullying.

The study suggests that an excessive focus on tasks, rigid hierarchies, or insufficient work–life balance policies can lead to perceptions of WB, whereas excessive employee focus and a low hierarchy might enable WB perpetration.

The findings align with social exchange theories, indicating that negative behaviours arise when employees feel unsupported by their organisations. A key insight is that a balance of high job demands and resources can prevent negative behaviours and contribute to employee well-being, supporting flow theory. Interestingly, a very small group of perpetrators was present even in these balanced environments, exhibiting negative behaviours and presenting a challenge for future researchers and practitioners to address.

## Figures and Tables

**Figure 1 behavsci-14-00751-f001:**
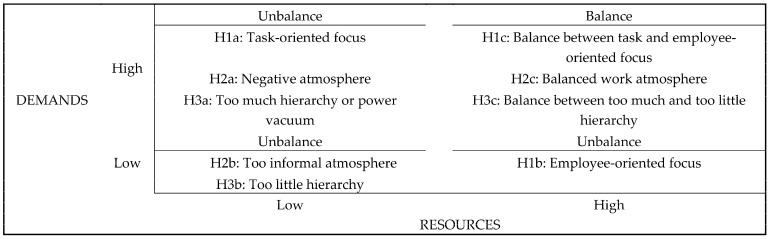
Hypotheses derived from the proposed model on the combination of the JD–R model [12] and the three-way model [14].

**Figure 2 behavsci-14-00751-f002:**
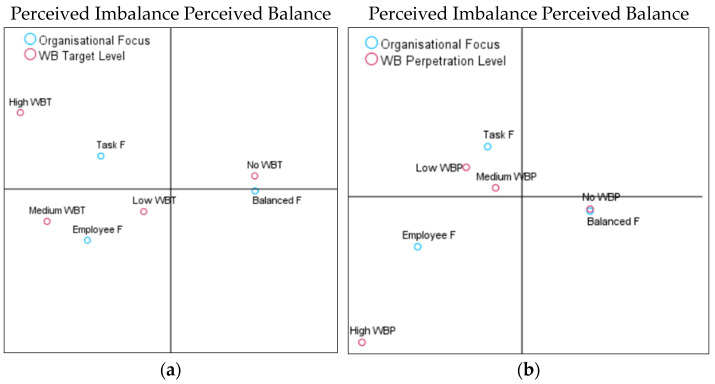
(**a**). Plot of category points on organisational focus and WB target. (**b**). Plot of category points on organisational focus and WB perpetration.

**Figure 3 behavsci-14-00751-f003:**
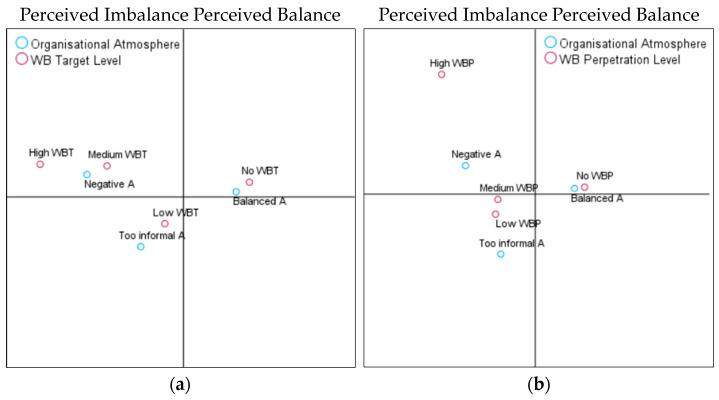
(**a**). Organisational atmosphere and WB target. (**b**). Organisational atmosphere and WB perpetration.

**Figure 4 behavsci-14-00751-f004:**
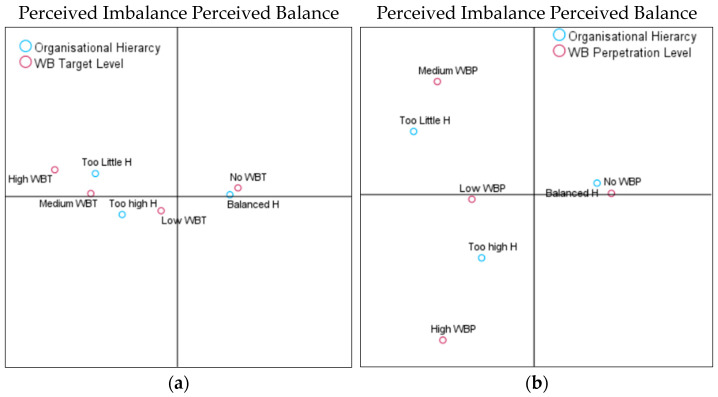
(**a**). Organisational hierarchy and WB target. (**b**). Organisational hierarchy and WB perpetration.

**Figure 5 behavsci-14-00751-f005:**
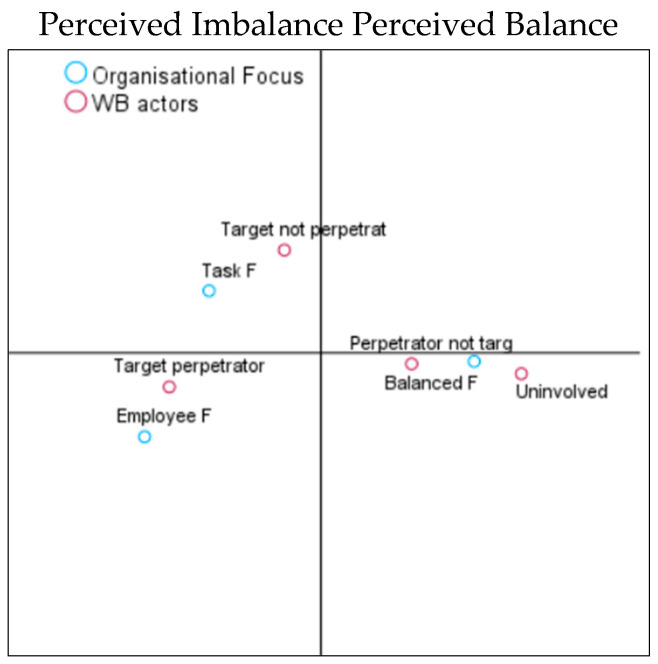
Organisational focus and WB actors.

**Figure 6 behavsci-14-00751-f006:**
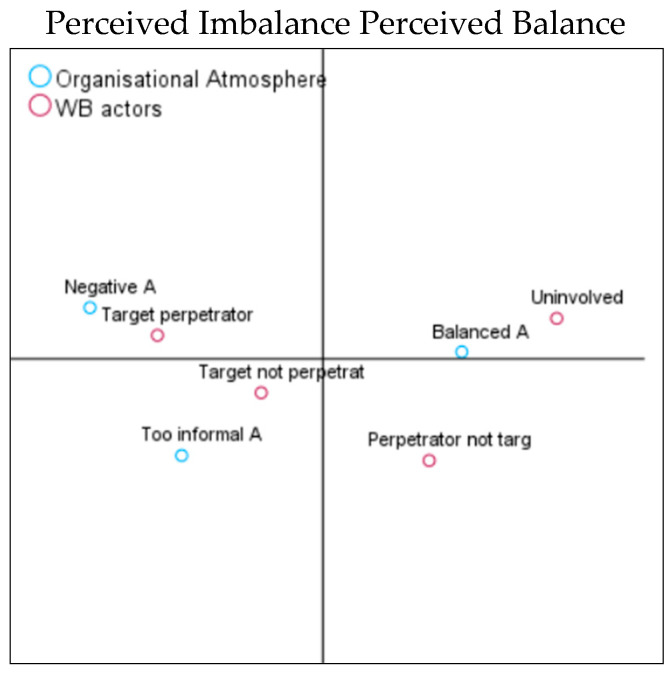
Organisational atmosphere and WB actors.

**Figure 7 behavsci-14-00751-f007:**
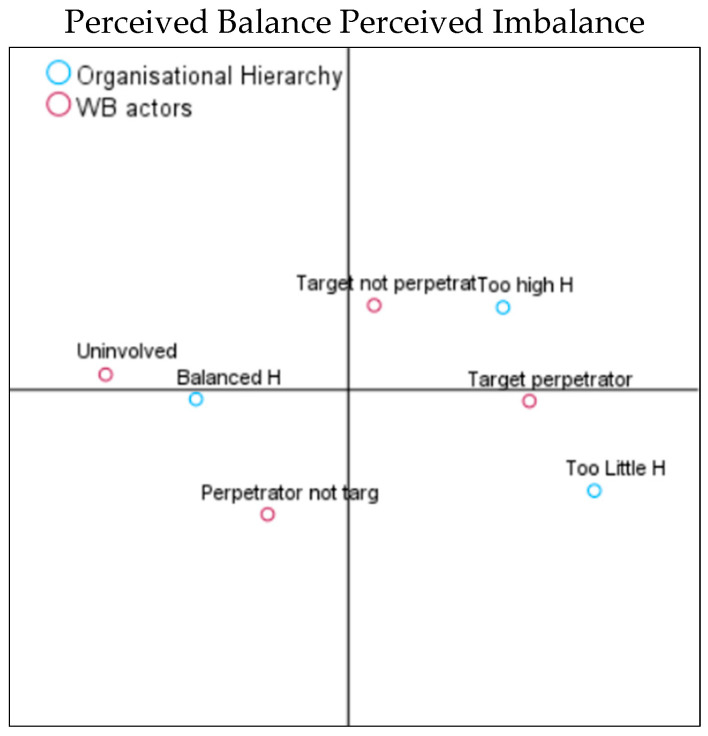
Organisational hierarchy and WB actors.

**Table 1 behavsci-14-00751-t001:** Sociodemographic characteristics of participants at baseline.

Baseline Characteristics	n	%		n	%
Gender			Supervisor		
Males	404	38.7	Not a supervisor	872	83.5
Females	640	61.3	Supervisor	172	16.5
Sector			Civil Status		
Education	257	24.6	Single	431	41.3
Industry	96	9.2	Married/Living together	523	50.1
Trade	151	14.5	Separated/divorced	80	7.7
Services	540	51.7	Widowed	10	1.0
Education			Income		
No studies	14	1.3	Equal or less than €10k	221	21.2
Basic	123	11.8	€10,001–€20,000	338	32.4
Secondary	409	39.2	€20,001–€30,000	303	29.0
Diploma	211	20.2	€30,001–€40,000	117	11.2
Undergraduate	219	21.0	€40,001–€50,000	32	3.1
Postgraduate	68	6.5	More than €50,000	33	3.2
Contract					
No permanent contract	304	29.1			
Permanent contract	740	70.9			

Note: n = 1044.

**Table 2 behavsci-14-00751-t002:** Classification of organisational dimensions and adjectives.

Organisational Dimensions	Continuum	Example Adjectives
1. Organisational Focus	Task-Focused	Exploitative, obsolete, statistical
	Balanced Focus	Organised, participative, supportive
	Employee-Focused	Unstructured, disorganised, chaotic
2. Organisational Atmosphere	Hostile or Negative	Controlling, manipulative, inhumane
	Balanced or Positive	Amiable, respectful, empathetic
	Too Informal	Overwhelmed, unmotivated, suffocating
3. Organisational Hierarchy	Too Much	Authoritarian, inefficient, dictatorial
	Balanced Hierarchy	Hierarchical, cheerful, coherent
	Too Little	Uncoordinated, little prepared, unclear

**Table 3 behavsci-14-00751-t003:** Descriptives and Pearson correlations.

Variables		Mean	SD	1	2	3	4	5
1	Age	35.43	10.91	-				
2	Gender	1.61	0.49	0.06	-			
3	Supervisor	1.16	0.37	0.10 **	−0.08 *	-		
4	WB Target Score	0.24	0.41	−0.07 *	−0.04	0.01	-	
5	WB Perpetration Score	0.22	0.39	−0.06	−0.08 *	−0.00	0.52 **	-

Notes: Gender 1 = males, 2 = females; Supervisor 1 = No, 2 = Yes, * *p* < 0.05; ** *p* < 0.01.

**Table 4 behavsci-14-00751-t004:** Frequencies and percentages of WB experience levels.

	WB Perpetration Level									
WB Target Level	No WBP	Low WBP	Medium WBP	High WBP	Total	No WBP	Low WBP	Medium WBP	High WBP	Total
No WBT	295	131	7	12	445	28.3%	12.5%	0.7%	1.1%	42.6%
Low WBT	173	228	19	15	435	16.6%	21.8%	1.8%	1.4%	41.7%
Medium WBT	12	43	17	4	76	1.1%	4.1%	1.6%	0.4%	7.3%
High WBT	26	28	21	13	88	2.5%	2.7%	2.0%	1.2%	8.4%
Total	506	430	64	44	1044	48.5%	41.2%	6.1%	4.2%	100.0%

Note: WBT: Workplace bullying target, WBP: Workplace bullying perpetrator.

**Table 5 behavsci-14-00751-t005:** Participant categories and acts.

Categories	N	%
Target not a perpetrator	211	20.2
Target perpetrator	388	37.2
Perpetrator not a target	150	14.4
Uninvolved	295	28.3
Total	1044	100.0

**Table 6 behavsci-14-00751-t006:** Correspondence table on organisational focus versus WB actor categories.

WB Categories	Task Focus	Balanced Focus	Employee Focus	Total	Task Focus	Balanced Focus	Employee Focus	Total
Target not a perpetrator	55	66	29	150	7.3%	8.8%	3.9%	20.0%
Target perpetrator	112	107	75	294	15.0%	14.3%	10.0%	39.3%
Perpetrator not a target	30	54	17	101	4.0%	7.2%	2.3%	13.5%
Uninvolved	52	125	27	204	6.9%	16.7%	3.6%	27.2%
Total	249	352	148	749	33.2%	47.0%	19.8%	100.0%

Note: In total, 749 employees gave adjectives on organisational focus among 1044 participants.

**Table 7 behavsci-14-00751-t007:** Correspondence table on organisational atmosphere versus WB actor categories.

WB Categories	Negative A	Balanced A	Too Informal A	Total	Negative A	Balanced A	Too Informal A	Total
Target not a perpetrator	40	78	28	146	5.5%	10.6%	3.8%	19.9%
Target perpetrator	98	132	60	290	13.4%	18.0%	8.2%	39.6%
Perpetrator not a target	17	65	16	98	2.3%	8.9%	2.2%	13.4%
Uninvolved	24	153	22	199	3.3%	20.9%	3.0%	27.1%
Total	179	428	126	733	24.4%	58.4%	17.2%	100.0%

Note: In total, 733 employees indicated adjectives on organisational atmosphere among 1044 participants; A = Atmosphere.

**Table 8 behavsci-14-00751-t008:** Correspondence table on organisational hierarchy versus WB actor categories.

WB Categories	Too Little H	Balanced H	Too High H	Total	Too Little H	Balanced H	Too High H	Total
Target not a perpetrator	25	79	44	148	3.4%	10.8%	6.0%	20.2%
Target perpetrator	68	124	96	288	9.3%	16.9%	13.1%	39.3%
Perpetrator not a target	16	61	22	99	2.2%	8.3%	3.0%	13.5%
Uninvolved	17	143	38	198	2.3%	19.5%	5.2%	27.0%
Total	126	407	200	733	17.2%	55.5%	27%	100.0%

Note: In total, 733 employees indicated adjectives on organisational hierarchy among 1044 participants; H = Hierarchy.

## Data Availability

Data are available at 10.6084/m9.figshare.26185757.

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
