# Peer review of "Imbalance between Employees and the Organisational Context: A Catalyst for Workplace Bullying Behaviours in Both Targets and Perpetrators"

_behavsci, 2024, doi:10.3390/bs14090751_

Round 1

Reviewer 1 Report

Comments and Suggestions for Authors

General considerations:

I believe that the subject of this research is relevant, and the authors use an interesting approach to the subject. The paper's title, abstract, and main sections are consistent with the data. The study's objectives are clear, and the method is adequate. The data presented support the conclusions.

I have only a few small suggestions for the authors:

Correspondence analysis on hypothesis

Page 10, 11 and 12: figures 2a and 2b, and figures 3a and 3b, and figures 4a and 4b I think you should try to put the legends under each figure, it would be easier for the reader.

After these small changes, I believe the paper is ready to be published.

Author Response

Thank you very much for your review. All of the indicated formatting was done.

We are grateful for your time.

Best regards

Reviewer 2 Report

Comments and Suggestions for Authors

The article presents an empirical study that aims to gather empirical evidence (with a large sample) to support a conceptual model, which aims to frame the phenomenon of workplace bullying with contextual variables in organisations. The study uses a range of methodologies, with the capacity to construct a more complex view of the phenomenon, and therefore represents a relevant contribution to the production of knowledge about workplace bullying in organisations.

For these two reasons, I recommend the publication, with a few minor corrections:

1) Line 36, although the acronym WB (workplace bullying) has been explained in the abstract, it would be beneficial to do so again the first time it is used in the text.

2) Lines 51-53, I suggest aggregating sentences that deal with the same subject in the same paragraph, especially when the first sentence is short (1 line and a bit more). Please consider this suggestion.

3) In lines 56, 516, 534,539-540, the font and the size of the characters need to be corrected. Please rectify. 

4) The information on guarantees of anonymity is repeated twice (Lines 250 and 288). Please rectify.

5) The use of decimal places in the numerical values (percentages) in the Tables should be standardised. In Table 6 (no decimal values are used in the %, while they are used in the other tables). Please correct.

6) The conclusion (Lines 618-623) is short. Considering the originality of the study and the potential of its contribution to the production of knowledge about the phenomenon, the conclusion deserved more development, emphasising the added value generated by the conceptual model and the methodology used in data collection.

7) In the references, there are several corrections to be made:

7.1) Remove the use of capital letters in the words of the article title (except for the first one), e.g. reference 1, 8,19,22... Please correct.

7.2) Remove the name of the publisher's locality from references 15, 18, 23, 34 48... Please correct.

Author Response

The article presents an empirical study that aims to gather empirical evidence (with a large sample) to support a conceptual model, which aims to frame the phenomenon of workplace bullying with contextual variables in organisations. The study uses a range of methodologies, with the capacity to construct a more complex view of the phenomenon, and therefore represents a relevant contribution to the production of knowledge about workplace bullying in organisations.

For these two reasons, I recommend the publication, with a few minor corrections:

1) Line 36, although the acronym WB (workplace bullying) has been explained in the abstract, it would be beneficial to do so again the first time it is used in the text. – Formatting is done.

2) Lines 51-53, I suggest aggregating sentences that deal with the same subject in the same paragraph, especially when the first sentence is short (1 line and a bit more). Please consider this suggestion. – – Formatting is done.

3) In lines 56, 516, 534,539-540, the font and the size of the characters need to be corrected. Please rectify. - – Formatting is done.

4) The information on guarantees of anonymity is repeated twice (Lines 250 and 288). Please rectify. - Line 250 is deleted.

5) The use of decimal places in the numerical values (percentages) in the Tables should be standardised. In Table 6 (no decimal values are used in the %, while they are used in the other tables). Please correct.- Corrected.

6) The conclusion (Lines 618-623) is short. Considering the originality of the study and the potential of its contribution to the production of knowledge about the phenomenon, the conclusion deserved more development, emphasising the added value generated by the conceptual model and the methodology used in data collection.- Thank you for your comment. We revised the conclusion as below.

“The study advances the understanding of organizational environments and their role as enablers of WB experiences by examining them from both the perpetrators' and targets' perspectives, addressing a gap in the literature with an original methodological approach.

An organizational setting that maintains a balanced focus on both its employees and tasks—where the organization is well-organized, coherent, supportive, empathetic, and respectful of human relations—energizes and motivates employees, leading to high performance even in the face of significant work demands. The research challenges the narrow focus on high demands causing strain, showing that high demands can be beneficial for employees when supported by adequate resources. Such balanced organizational contexts are associated with the absence of WB. Conversely, organizational settings that are disorganized, unmotivating, or controlling and exploitative of their employees miss the opportunity to unlock their employees’ potential. Imbalanced contexts lead to higher reports of WB, while balanced environments show no involvement in bullying.

The study suggests that an excessive focus on tasks, rigid hierarchies, or insufficient work-life balance policies can lead to perceptions of WB, whereas excessive employee focus and low hierarchy might enable WB perpetration.

The findings align with social exchange theories, indicating that negative behaviors arise when employees feel unsupported by their organizations. A key insight is that a balance of high job demands and resources can prevent negative behaviors and contribute to employee well-being, supporting flow theory. Interestingly, a very small group of perpetrators was present even in these balanced environments, exhibiting negative behaviors and presenting a challenge for future researchers and practitioners to address.”

7) In the references, there are several corrections to be made:

7.1) Remove the use of capital letters in the words of the article title (except for the first one), e.g. reference 1, 8,19,22... Please correct. – Formatting is done.

7.2) Remove the name of the publisher's locality from references 15, 18, 23, 34 48... Please correct. -– Formatting is done.

Thank you very much for your review. The indicated changes in formatting were done, conclusions section is expanded.  

We are grateful for your time.

Best regards